# Advancing Neuromorphic Computing Algorithms and Systems with NeuroBench

**Jason Yik**[1]**, Charlotte Frenkel**[2]**, Vijay Janapa Reddi**[1]

[1] Harvard University [2] TU Delft

## Abstract

Neuromorphic research aims to approach the capabilities and efficiency of the brain by developing algorithm and hardware architectures which adopt mechanical features of biological computing. As improvements in computing speeds slow and new workloads scale towards untenable costs in conventional approaches, neuromorphic computing offers a novel paradigm for neural network intelligence and hardware architectures which may realize a next-generation of computing performance. Such a paradigm optimizes for different goals than conventional computing systems, and therefore novel benchmark assessment and methods are necessary for technological progress. In this article, we present NeuroBench, a benchmark framework for neuromorphic computing which has engaged over 100 researchers from more than 50 institutions across academia and industry. NeuroBench provides novel benchmark tasks and common, extensible tools for evaluation, unifying the space of neuromorphic research. Results from NeuroBench benchmarks and challenges highlight the strengths of neuromorphic algorithms and systems for sparse, event-driven, and energy-efficient machine learning.

## 1 Introduction

The brain, a computer evolved by nature, boasts intelligence capabilities and a minuscule energy consumption that are both far outside the realm of existing computing technologies. Approximating between consumed calories and mechanical power, the brain operates using a power budget in the order of 20W (1), which could not begin to power any general-purpose AI system. For reference, one NVIDIA H100 has a thermal design power (TDP) of 700W (2). As the energy expenditure of conventional AI computing accelerates without foreseeable end (3), it is prudent to study techniques towards more energy-efficient AI.

Neuromorphic research aims to approach biological capability and efficiency by developing algorithms and hardware systems which incorporate mechanical features of biological computation, such as sparsity, decentralized parallelism, and event-driven operation. One key algorithmic approach in neuromorphic AI research is the spiking neural network (SNN). SNNs feature stateful, binary-activation, sparsely-activated neuron models, similar to biological neurons and unlike conventional artificial neural network (ANN) neuron models, which are typically stateless, continuous-activation, and densely-activated. Such features enable SNNs to be used for AI tasks with far lower effective operational costs than densely-calculated ANNs, while maintaining comparable fidelity. Algorithmic research in SNNs continues to close the accuracy gap with ANNs across domains such as computer vision (4) and language processing (5), demonstrating high task accuracy and theoretical efficiency.

However, so far, SNNs have not yet been widely recognized on their potential for capable, highly efficient AI. This is because fundamentally, SNNs cannot realize theoretical efficiency on the conventional hardware systems that power current AI, such as GPUs. Such conventional systems are specialized to full-utilization for dense compute operation, but to realize efficiency utilizing the highly dynamic sparsity in SNNs, a different approach is needed. Thus, new types of neuromorphic

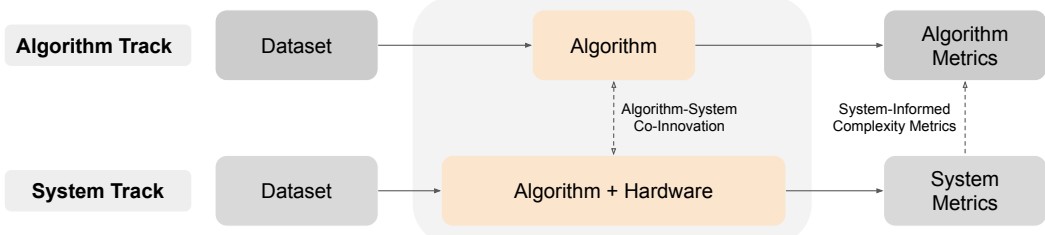

Figure 1: The dual-track structure of NeuroBench. (19).

hardware architectures (e.g., Intel Loihi (6), SpiNNaker (7), Synsense Xylo (8), BrainScaleS (9), to name a few) are being developed for co-designed algorithm+hardware systems which can exploit the sparse, event-driven features of neuromorphic algorithms for efficient deployed performance.

In order for neuromorphic research to advance and realize next-gen performance and efficiency, standardized benchmarks must be used to objectively quantify and compare promising solutions. The neuromorphic approach represents a full-stack paradigm, which in turn warrants specialized benchmark tasks and measurement methodology. Classic deep learning benchmarks such as MNIST (10), CIFAR (11), and ImageNet (12) have been commonly applied to neuromorphic research. However, the tasks focus on offline, batch processing of image frames, while neuromorphic algorithms and hardware are geared more towards real-time operation on natively temporal data, such as video or event-camera data.

Furthermore, when measuring benchmark performance, conventional benchmarks will have a primary, often exclusive, focus on algorithmic correctness (accuracy) and system runtime (13; 14; 15). In contrast, neuromorphic approaches are designed with efficiency concerns as a primary objective and must be benchmarked accordingly. Conventional benchmarks may use measures such as floating-point operations (FLOPs) to quantify compute costs, but this metric is geared towards dense compute throughput, and not sparse or event-based operation.

While different benchmark tasks are used which better align with neuromorphic characteristics such as sparse, event-based time series data (e.g. N-MNIST (16), Spiking Heidelberg Digits (17), and DVS Gesture (18)), standardized evaluation methodology for analyzing compute costs at the algorithmic and system levels are necessary. A complete benchmarking framework must unite benchmark tasks, which concern task accuracy, with measurements that account for the final efficiency and cost of a solution.

To this end, this article presents recent advancements with the NeuroBench benchmark framework (19). NeuroBench is a collaboratively-designed initiative engaging over 100 participants across over 50 academic and industry institutions with interests in neuromorphic research. NeuroBench delivers a platform for novel benchmarks for both neuromorphic algorithms and systems, which are unified by a common set of metrics and measurement methodologies. It provides an open-source, extensible benchmark harness tool which enables simple solution profiling and benchmark comparisons, and encourages open research code and cross-software integrations to accelerate research development. At a high level, NeuroBench aims to inclusively apply for benchmarking of novel neuromorphic solutions, actionably enable researchers to implement benchmarks, and iteratively expand with extensible and practical design principles.

This article overviews the NeuroBench framework and recent benchmarks which have been demonstrated. Further up-to-date information can be found at the project website: neurobench.ai.

## 2   The NeuroBench Framework

Although neuromorphic hardware designs have been developed and even are becoming viable commercial products, hardware platforms still lack standardized system interfaces and are not available off-the-shelf for research. As such, current neuromorphic algorithm research, such as machine learning with SNNs, is conducted for the most part on conventional platforms like GPUs,

| Task | Dataset | Task description |
|------|---------|------------------|
| Keyword FSCIL | MSWC (20) | Few-shot, continual learning of keyword classes. |
| Event Camera Detection | Prophesee 1MP Automotive (21) | Detecting automotive objects from event camera video. |
| Primate Motor Decoding | Primate Reaching (22) | Predicting fingertip velocity from cortical recordings. |
| Time Series Forecasting | Mackey-Glass (23) | Autoregressive modeling of chaotic functions. |

Table 1: NeuroBench algorithm track novel benchmarks.

which are non-optimal for final deployment. Steadily, neuromorphic algorithms are being tuned and deployed for neuromorphic hardware, forming complete systems.

To facilitate and bridge between research, the NeuroBench framework is composed of both an algorithm track and a system track of benchmarks (Figure 1). The algorithm track is intended for agile, hardware-independent evaluation, and defines general compute complexity metrics in order to analyze solution costs. Meanwhile, the system track is for fully deployed solutions, defining standard protocols to measure real-world speed and efficiency of neuromorphic hardware platforms. Between the two tracks, features of successful existing solutions will influence the requirements and configurations of future research. Furthermore, the interactions between algorithm track complexity metrics and deployed measurements from the system track will inform algorithmic profiling and expedite prototyping and exploration.

## 2.1 Algorithm Track

### 2.1.1 Metrics

The algorithm track establishes primary metrics which are generally relevant to different types of solutions, including ANNs and SNNs. Firstly, there are *correctness* metrics, which measure the quality of the model predictions on the particular task. Next, there are *complexity* metrics, which measure the compute costs of the algorithm, profiling it to offer intuition into deployed performance and efficiency on specialized neuromorphic hardware. The complexity metrics include:

- Footprint, accounting for data buffering, neuron states, weights
- Connection sparsity of synaptic weights, which accounts for pruning and deliberately sparse connectivity designs
- Activation sparsity, the frequency of neurons outputting no (zero) activations
- Synaptic operations, divided into Dense (includes all zero ops), Effective Multiply-and-Accumulates (for continuous-valued activations, excluding zero ops), and Accumulates (for binary activations in SNNs, excluding zero ops)
- Model execution rate, the rate that the model forward pass is executed with respect to the data time

The complexity metrics are measured independently of the underlying hardware and therefore do not explicitly correlate with post-deployment latency or energy consumption. However, they provide valuable insight into algorithm performance and resource requirements, enabling high-level comparison and facilitating prototyping. For instance, the execution rate and number of synaptic operations can be taken together to estimate the speed and dynamic power of a model deployed to certain hardware, and the footprint and connection sparsity can be used to proxy hardware resource utilization.

### 2.1.2 Benchmarks

The above metrics can be generally applied across benchmark tasks, and NeuroBench introduces baseline implementations on four novel tasks, which are summarized in Table 1. The benchmarks cover a diverse range of applications of interest to neuromorphic algorithms, including continual learning, event camera vision, motor cortical decoding, and sequence forecasting.

```
1   import torch
2   from torch.utils.data import DataLoader
3   from neurobench.datasets import SpeechCommands
4   from neurobench.preprocessing import S2SPreProcessor
5   from neurobench.postprocessing import choose_max_count
6   from neurobench.models import SNNTorchModel # model wrapper for snntorch
7   from neurobench.benchmarks import Benchmark
8
9   dataset = SpeechCommands(path, subset="testing")
10  loader = DataLoader(test_set, batch_size, ...)
11
12  net = torch.load(...) # load a trained snntorch model
13  model = SNNTorchModel(net)
14
15  pre = [S2SPreProcessor()]
16  post = [choose_max_count]
17  static_metrics = ["footprint", "connection_sparsity"]
18  workload_metrics = ["classification_accuracy", "synaptic_operations"]
19
20  benchmark = Benchmark(model, loader, pre, post, [static_metrics, workload_metrics])
21  results = benchmark.run()
```

Listing 1: Example user-level interface for benchmarking an SNN with the NeuroBench code harness.

### 2.1.3  Code Harness

The algorithm track benchmarks and metrics are all encapsulated within a common, open-source harness tool. Given a trained model and an evaluation dataset, the harness automatically tests the model on the dataset, and calculates metrics at runtime. The harness uses a simple interface and is compatible with all PyTorch and PyTorch-based libraries for datasets and machine learning, and included tutorials already demonstrate integration with popular libraries used in neuromorphic research (24; 25; 26). Listing 1 shows an example user-level interface for the code harness.

While the current code harness has made a first step towards research tool integration, further extensions can increase the inclusivity of the standard tooling. For example, PyTorch-based frameworks mainly cover discrete-time execution models which are not compatible with continuous-time models that are essential for algorithms intended for analog or mixed-signal hardware. In addition, more specific metrics such as robustness to noise and maximum fanout may be important when considering such algorithms. The simple, standard interfaces of the harness have been designed to facilitate extensions, including custom metric definitions and integration with tools like NIR (27), an intermediate representation which bridges between discrete- and continuous-time simulation libraries.

### 2.1.4  Results

| Model | $R^2$ | Footprint (bytes) | SynOps | | |
| --- | --- | --- | --- | --- | --- |
| | | | Dense | Eff_MACs | Eff_ACs |
| ANN Baseline | 0.5755 | 27160 | 6236 | 4970 | 0 |
| SNN Baseline | 0.5805 | 29248 | 7300 | 0 | 413 |
| AEGRU (28) | 0.6982 | 45520 | 54283 | 25316 | 0 |
| RSNN-L (29) | 0.6978 | 4833360 | 1206272 | 0 | 42003 |
| RSNN-S (29) | 0.6604 | 27144 | 13440 | 0 | 304 |
| ConvGRU (30) | 0.6209 | 26568 | 4947 | 627 | 247 |

Table 2: Results for nonhuman-primate motor decoding task. The top two rows are NeuroBench baselines (19), and the four lower results are from the recent BioCAS Grand Challenge (31).

Results are highlighted from the NeuroBench non-human primate motor decoding task in Table 2. The task is to decode motor cortical signals from monkeys who are engaged in reaching tasks - given the neural data sequence, predict the sequence of fingertip velocities. The task lends to BMI applications,

which may be subject to extreme area and energy constraints. The NeuroBench baselines (Table upper rows) demonstrated how a small SNN outperforms a small ANN, achieving higher $R^2$ (0.5805 vs 0.5755) while operating with fewer effective non-zero operations (413 vs 4970), which further are accumulations and would not require multiplication hardware.

Recently, this task was used as a Grand Challenge with the IEEE BioCAS conference (31), inviting researchers to compete in achieving high correctness with low operational complexity. The four winning submissions (Table lower rows) cover both recurrent ANN (GRU) approaches and recurrent SNNs. The training effectiveness to produce higher $R^2$ is impressive (up to 0.6982), and even more notable are the performance and low footprint and operations of the large and small SNNs (0.6604 with only 304 effective synaptic operations). The NeuroBench algorithm track benchmark and harness has provided a platform to highlight state-of-the-art techniques for neuromorphic algorithms, compare them against conventional approaches, and motivate further research progress.

## 2.2 System Track

While the NeuroBench algorithm track is meant for hardware-independent profiling and prototyping, the system track targets benchmarking full neuromorphic platforms. As the development of systems involves far more engineering effort including algorithm, hardware, and software development, the initial focus of the system track is to provide a unified platform for which system developers can showcase their results head-to-head, in contrast to providing tools and methods for profiling as in the algorithm track.

Deployed metrics are compared between novel neuromorphic designs, as well as against conventional platform baselines, in order to demonstrate where advantages and strengths lie. The system track also offers a framework for documenting the methods used for designing and measuring different architectures. By transparently sharing and documenting, the system track intends to drive neuromorphic hardware towards mature standardization.

### 2.2.1 Metrics and Measurement

The system track benchmarks assess platforms at the task level, using metrics such as execution time and throughput, and power and energy consumption. While the metrics are straightforward, fairly measuring varying hardware platforms presents a large challenge. For one, neuromorphic platforms are implemented at various solution scales, from single-chip, ultra-efficient edge devices (32) up to multi-board server machines containing hundreds of chips (33). Furthermore, neuromorphic devices often utilize analog or mixed-signal technology (8)(9), which have significantly different characteristics than fully-digital platforms. Also, the state of many neuromorphic systems may not be a mature, standalone machine, and thus development and experimental hardware may populate the system which shouldn't reasonably be included in measurements. All of the challenges make a singular, fully-consistent measurement methodology that everyone can strictly apply currently infeasible.

In order to actionably benchmark neuromorphic systems, in the absence of consistency the NeuroBench system track instead favors transparency. Based on features of existing neuromorphic systems, the track offers guidelines for measurement protocols and reporting documentation to contextualize system track results. Transparent documentation provides the foundation for a shared methodology between varying systems, and it allows for wide system exploration with holistic analysis. Using the current generation of reports, future generations of system design will have development guides to improve system consistency, lending to both system maturation and standardization across the field.

### 2.2.2 Benchmarks

Table 3 lists the two benchmarks of the initial version of the NeuroBench system track. The tasks cover both edge classification, as well as mobile- and server-scale graph optimization, which was included to demonstrate that the principles of neuromorphic hardware can lend efficiency not only to machine learning, but also to other applications with sparse and irregular workloads, such as graph optimization. While the algorithm track's initial benchmark suite features challenging research tasks, the initial system benchmark suite aligns with the scale and intended use cases of existing neuromorphic systems, and the two suites are intended to coalesce as research and systems mature.

| Task | Dataset | Task description |
|------|---------|------------------|
| Acoustic Scene Classification | DCASE (34) | Classifying environments from ambient noise. |
| QUBO | Graphs spanning size, density | Optimization to find maximal independent sets in graphs. |

Table 3: NeuroBench system track benchmarks.

### 2.2.3 Results

| Baseline | | Accuracy | Execution Time (ms) | Power (mW) | Energy (mJ/inf) |
|----------|---|----------|---------------------|------------|-----------------|
| CPU *(system-wise)* | *pre-process* | 79.64% | 43 | 21.32 | 0.917 |
| | *inference* | | 45 | 20.75 | 0.934 |
| Xylo *(component-wise)* | *pre-process\** | 79.90% | - | 0.015* | 0.015* |
| | *inference* | | 84 | 0.341 | 0.028 |

Table 4: NeuroBench initial results for the acoustic scene classification task. Xylo pre-processing (denoted by astrisk *) utilizes an analog module, and power is measured in real-time using sensor data, while all other modules measure timing and power with digital data. Power measures difference between floor power and execution power. More details available in (19; 35).

System track results from the acoustic scene classification benchmark are highlighted in Table 4. The Synsense Xylo (8) is compared against an Arduino CPU microcontroller system, and uses $60.9\times$ less inference power and $33.4\times$ less inference energy. The comparison demonstrates benchmark consistency challenges due to the variability of system configurations: the Xylo uses an analog pre-processing module operating in real-time on sensor data, while all other measured components use digital data. Following the NeuroBench focus on transparency, benchmark reporting provides in-depth details on the Xylo system implementation (35).

## 3 Opportunities

Standard benchmark tasks and measurement are necessary to compare between solutions and determine successful features, and are thus vital for technological progress. NeuroBench is an effort which offers this common benchmarking platform, both for algorithmic prototyping and profiling and for fully deployed systems. Many benchmarks have shown that neuromorphic designs can offer advantages over conventional technology, and now NeuroBench allows these designs to be directly contrasted under a common set of rules and tooling.

As more results are collected with both the algorithm and system benchmark tracks, neuromorphic researchers will be able to find task leaderboards, providing a target for new design metrics and offering a streamlined location to review state-of-the-art research and build on prior work. In addition, the open-source code harness of the algorithm track and transparent documentation guidelines of the system track will enable researchers to share, reproduce, and extend on each other's work, driving research discovery by reducing engineering workloads.

The near-term next steps with NeuroBench offer exciting opportunities for neuromorphic research. Further benchmark challenges can make use of the benchmarks and tooling in order to incentivize innovation for different applications. The framework is grounded on iterative extension, and novel benchmark tasks and metric definitions are being proposed and developed. The divide between the algorithm and system tracks will narrow as complexity analyses from the former side are collected and understood in the context of deployed hardware performance in the latter, paving the way for optimal system hardware-software co-design.

Researchers interested in joining the community, organizing benchmark challenges, extending tooling, and staying up-to-date can follow NeuroBench from the project website: `neurobench.ai`.

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
