# OpenReview forum: "Advancing Neuromorphic Computing Algorithms and Systems with NeuroBench"
_NeurIPS.cc/2024/Workshop/MLNCP — MLNCP Poster_

### Official Review · Reviewer_YRHc · 2024-09-21
**Work in Progress with Limited Significance**

**Rating:** 6
**Confidence:** 5

**Review:**

The paper addresses an important issue relevant to the workshop which is a lack of standardized methods and benchmarks for neuromorphic hardware implementing spiking neural networks. The authors approach this issue in a systematic way and correctly recognize challenges such as the need to distinguish between algorithmic properties (e.g., memory requirements) and the properties of hardware implementations that ultimately determine the performance of neuromorphic systems. However, the described work is clearly in progress and, at this point, it is questionable whether the described methods significantly advance the state-of-the-art or are useful to serve the intended purpose of providing standard benchmarking methodology for neuromorphic hardware in general.

In my opinion, the authors should either reduce the scope of their work (e.g., to characterizing implementations based on PyTorch and digital neuromorphic hardware) or attempt to develop evaluation metrics that are generally applicable and defer publication until the project, specifically the "system track", has matured beyond statements of intent.

Constructive criticism:
* The manuscript only discusses evaluating inference. However, the bottleneck of model size in ML is given by the energy and memory demands during training. The authors should discuss or develop benchmarks or metrics for the costs of model training and how these might differ between classical and neuromorphic implementations.
* The "algorithm track" seems to be geared towards PyTorch-based SNN implementations. These typically are discrete-time and formally equivalent to recurrent binary neural networks/multi-layer Perceptrons. Authors should discuss how this blurs the line between traditional machine learning and neuromorphic computing.
* The "algorithm track" seems to be inapplicable to any neuromorphic algorithm implementation that is not based on PyTorch. The authors should discuss how analog or mixed-signal hardware could fit into the described framework. Can the provided "code harness" be used with SpiNNaker2, Loihi, BrainScaleS, ... or would this require re-implementing the models using PyTorch?
* Leaving aside the "code harness" that automatically computes complexity metrics for PyTorch implementations, what is the novelty or scientific contribution of the algorithm track? The described metrics in 2.1.1 are standard and authors should consider constructing metrics that capture a trade-off between model performance and computational complexity.
* ANNs such as LLMs are typically quantized to lower precision arithmetic for inference and GPUs such as the H100 have native support for, for example, FP8 MAC. Authors should take this into account and discuss how this impacts how meaningful metrics such as the number of effective MAC operations are.
* The "system track" seems to currently only consist of guidelines encouraging transparency when profiling neuromorphic hardware. While such guidelines may be useful, the authors should describe them in the manuscript instead of a general mention.
* Why are the benchmark data sets different for both tracks? For example, it seems highly relevant to characterize task-level performance for event-based vision.
* GPUs (e.g., the mentioned H100) and other ML accelerators (e.g., Cerebras) have features to accelerate sparse matrix multiplication. Authors however claim "Such conventional systems are specialized to full-utilization for dense compute operation". Authors should discuss how neuromorphic hardware differs from accelerated, sparse operations on traditional hardware.
* Authors should mention the specific contributions of "over 100 researchers". Readers may suspect that the majority of these researchers did not make a substantial contribution to the project and feel that this number has no bearing on the relevance or validity of the presented research.
* Authors should either explain what they mean by "agile analytical models" (l. 205) or avoid the use of such vague phrases.
* Authors should support claims such as "Many benchmarks have shown that neuromorphic designs can beat conventional technology" (l. 190-191) using citations.

---

### Official Review · Reviewer_Jsw3 · 2024-10-01
**fundamental benchmark framework with huge effort**

**Rating:** 8
**Confidence:** 4

**Review:**

Quality: The overall quality should be better than the average line. This work filled the gap on the benchmark perspective for neuromorphic computing, which can be taken as a elephant in the room.

Clarity: The article was written with idiomatic English to convey the information. The structure is logical and rational. Readers can position to specific sub-topic with minor effort.

Originality: This benchmark framework is a original huge work engaged with over 100 researchers from more than 50 institutions. It is observed that part of the figures and tables were taken from their pre-release in arxiv (2304.04640).

Significance: This article clearly stated the necessity of the proposed benchmark framework, which brings additional value, such as showcase of performance difference between ANN and SNN solutions for the same task, transparent task leaderboard, and a platform to share and extend on each other's work, for the community of neuromorphic computing.

Comment 1: For the collected tasks in the algorithm track, is there a way to display the task difficulty with a figure of merit? Algorithms or models can be sequentially challenged with more and more difficult tasks to evaluate their scalability.

Comment 2: Table 1 and 3 are wider than the text width

---

### Decision · Program_Chairs · 2024-10-10

Accept (Poster)